# MADrive: Memory-Augmented Driving Scene Modeling

## Abstract

Recent advances in scene reconstruction have pushed toward highly realistic modeling of autonomous driving (AD) environments using 3D Gaussian splatting. However, the resulting reconstructions remain closely tied to the original observations and struggle to support photorealistic synthesis of significantly altered or novel driving scenarios. This work introduces MADrive, a memory-augmented reconstruction framework designed to extend the capabilities of existing scene reconstruction methods by replacing observed vehicles with visually similar 3D assets retrieved from a large-scale external memory bank. Specifically, we release MAD-Cars, a curated dataset of ∼70K 360° car videos captured in the wild and present a retrieval module that finds the most similar car instances in the memory bank, reconstructs the corresponding 3D assets from video, and integrates them into the target scene through orientation alignment and relighting. The resulting replacements provide complete multi-view representations of vehicles in the scene, enabling photorealistic synthesis of substantially altered configurations, as demonstrated in our experiments.

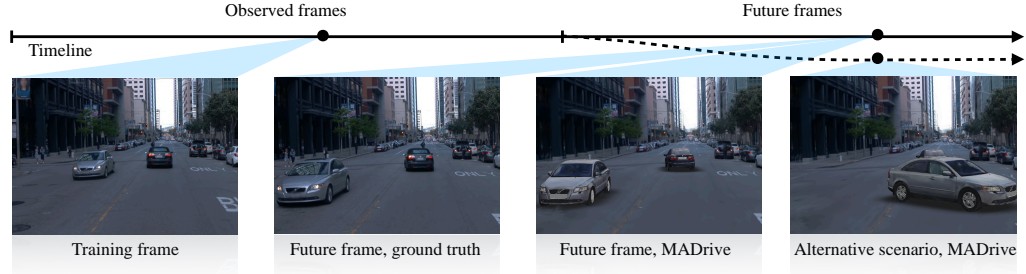

Figure 1: **MADrive** reconstructs a 3D driving scene from training frames (Left) and replaces partially observed vehicles in the scene with realistically reconstructed counterparts retrieved from **MAD-Cars**, our novel multi-view auto dataset. MADrive enables high-fidelity modeling of future scene views (Middle-left vs. Middle-right) and supports simulation of alternative scenarios, advancing novel-view synthesis in dynamic environments (Right).

## 1 Introduction

Autonomous driving (AD) is one of the key areas in computer vision, requiring extensive and costly data collection (Xiao et al., 2021; Sun et al., 2020; Geiger et al., 2012; Cabon et al., 2020; Caesar et al., 2020) to train accurate and robust perception and planning models (Bansal et al., 2018; Huang et al., 2024b; Gao et al., 2020). Driving simulators (Wang et al., 2023b; Zhou et al., 2024a; Yang et al., 2023a) aim at offering a powerful alternative by enabling the generation of highly realistic novel views and rare scenarios, especially safety-critical ones that are too dangerous or impractical to capture in the wild. Accurate modeling of such scenarios is essential to mitigate domain shift during training, which could otherwise lead to failures in deployments. When failures occur on real roads, it is important to "debug" the driving policy and "replay" the scenario to address the root cause.

Recent advances in multi-view reconstruction and novel view synthesis (Kerbl et al., 2023a; Yu et al., 2024; Kheradmand et al., 2024) provide a foundation for developing highly realistic driving simulators (Zhou et al., 2024a), designed to faithfully replicate real-world scenes. Such solutions can be used for real-time, controllable simulations that preserve the visual domain of real-world data, unlike game engine-based simulations, which often introduce significant domain shifts.

Modern driving scene reconstruction methods (Zhou et al., 2024b; Yan et al., 2024; Khan et al., 2024) have achieved impressive photo-realism in rendering the observed views, while also enabling the simulation of slight vehicle trajectory deviations, e.g., lane changes (Khan et al., 2024; Zhou et al., 2024b). However, since novel view synthesis is limited to the geometry observed in the data, existing reconstruction methods cannot reliably model vehicles beyond the observed views, which limits their usefulness for simulating alternative outcomes or reenacting failures in driving scenarios.

**Motivation.** In real-world driving, AD system failures often require human intervention. To diagnose such such failures, it is essential to reproduce the situation - modeling how events would have unfolded without intervention. Controllable reenactments not only help identify root causes, but also generate diverse training examples for robust AD. Therefore, we focus on controllable simulation of raw camera inputs to replicate frame sequences with similar surroundings and vehicles.

**Contributions.** To overcome the limitations of existing methods, we introduce MADRIVE, a memory-augmented reconstruction framework that integrates external 3D car models into captured driving scenes. To faithfully adapt these models to the surrounding scene, we further propose physically-based relighting and insertion techniques, resulting in visually consistent novel driving scene views.

Our method is motivated by the assumption that the variety of car models, types, and colors is relatively limited (Wikipedia contributors, 2024), making it feasible to build a dataset that covers the majority of cars typically seen on the road. To this end, we present a dataset containing $360°$ view sequences of $\sim 70,000$ cars, curated from online sale advertisements.

Furthermore, leveraging integrated high-fidelity car models enables a more challenging evaluation setting. Previous solutions for driving scene reconstruction mainly focus on replicating or editing unseen *intermediate* frames (Yan et al., 2024; Khan et al., 2024; Zhou et al., 2024b;a). Our work instead considers *driving scene extrapolation* — predicting the future appearance of vehicles based on a sequence of past views. We show that MADRIVE can faithfully render a diverse range of plausible vehicle trajectories, offering a foundation for simulation applications that model alternative outcomes of real driving scenarios. In summary, the contributions are as follows.

- We present MAD-CARS, **M**ulti-view **A**uto **D**ataset — a curated, large-scale collection of $360°$ car videos. It comprises $\sim 70,000$ car instances with diverse brands, models, colors, and lighting conditions, significantly expanding the scope of existing public multi-view car datasets.

- We propose MADRIVE– a **M**emory-**A**ugmented **D**riving scene reconstruction framework aimed at realistic synthesis of diverse and complex driving scenarios. Given sparse car views in the scene, MADRIVE retrieves similar vehicles from a car video database, reconstructs them into high-quality 3D assets, and naturally integrates them into the scene, replacing the original cars.

- We describe a novel evaluation setting for assessing driving scene reconstruction methods in significantly altered views, and show that MADRIVE produces more realistic renderings, as evidenced by reduced performance degradation in downstream perception tasks.

## 2 RELATED WORK

**Dynamic Urban Scene Reconstruction.** NeRFs (Mildenhall et al., 2020) can be used to model dynamic urban scenes. SUDS (Turki et al., 2023) uses a single network for dynamic actors, which limits the possibility of altering the behavior of the actors. EmerNeRF (Yang et al., 2024b) follows a similar idea to SUDS by decomposing the scene purely into static and dynamic components. NeuRAD (Tonderski et al., 2024) takes advantage of monocular or LiDAR-based 3D bounding box predictions and proposes a joint optimization of object poses during the reconstruction process. Although these methods produce reasonable results, they are still 1) limited to the high training cost and low rendering speed; or 2) do not address extrapolation of future vehicle appearance far beyond the original camera views. Recent dynamic 3D scene reconstruction methods increasingly adopt 3D Gaussian Splatting (Kerbl et al., 2023b) as an efficient and expressive scene representation (Yang et al., 2023b; Wu et al., 2024; Yang et al., 2024c; Chen et al., 2023). Several approaches [StreetGS (Yan et al., 2024), AutoSplat (Khan et al., 2024), HUGS (Zhou et al., 2024b)] adopt these methods to driving scene modeling and decompose the scenes into a static background and foreground vehicles, placed in the scene using 3D bounding boxes derived from tracking data. These methods also propose various modifications to improve driving scene reconstruction and novel view

synthesis. Both HUGS and AutoSplat represent the ground as a plane of 2D splats. HUGS further leverages additional information (optical flow and semantic segmentation) to guide splat optimization and introduces a method for realistic shadow placement. AutoSplat (Khan et al., 2024) improves car reconstruction from limited viewpoints by exploiting the bilateral symmetry of vehicles to augment side views and by employing more accurate splat initialization via an image-to-3D model (Pavllo et al., 2023). DrivingGaussian (Zhou et al., 2024c) uses composite dynamic Gaussian graph to handle multiple moving objects, individually reconstructing each object and restoring their accurate positions and occlusion relationships within the scene. OmniRe (Chen et al., 2025) leverages dynamic neural scene graphs based on Gaussian representations to unify the reconstruction of static backgrounds, driving vehicles, and non-rigidly moving dynamic actors, which enables human-centered simulations. Despite these advances, accurately reconstructing the full appearance of a vehicle in the scene, particularly from sparse or occluded views, remains a substantial challenge.

**3D Car Datasets.** Several public datasets provide 3D car assets. Early collections such as SRN-Car (Chang et al., 2015) and Objaverse-Car (Deitke et al., 2023) contain CAD models that deviate significantly from real-world cars in terms of texture realism and geometric details.

More recent efforts (Zhang et al., 2021; Du et al., 2024) have focused on real captured 3D car datasets. MVMC (Zhang et al., 2021) includes 576 cars, each with an average of 10 views. 3DReal-Car (Du et al., 2024) provides $2,500$ car instances, each with $\sim200$ dense high-resolution RGB-D views.

In contrast, MAD-CARS includes $\sim70,000$ 360° car videos at a comparable resolution and average number of views as 3DRealCar, thereby offering substantially greater generalization and diversity.

**Novel View Synthesis with External 3D Car Assets.** HUGSim (Zhou et al., 2024a) builds a closed-loop AD simulator by inserting 3D car models from 3DRealCars (Du et al., 2024). In contrast, we replace observed vehicles in real scenarios with retrieved counterparts, enabling extrapolation and rollout of actual driving situations.

Several approaches leverage CAD models for scene representation (Engelmann et al., 2017; Wang et al., 2023b; Uy et al., 2020; Avetisyan et al., 2019; Gümeli et al., 2022), but these assets often differ notably from real vehicles. To improve realism, some methods apply geometry tuning (Uy et al., 2020; Wang et al., 2023b; Engelmann et al., 2017), whereas UrbanCAD (Lu et al., 2024) retrieves similar CAD models and refines their textures and lighting to better match the scene while preserving CAD-level controllability. However, the obtained models still have a noticeable gap in realism and correspondence to actual cars.

Meanwhile, MADRIVE retrieves real cars instances from a large-scale database spanning diverse brands, models, materials, colors, and lighting conditions — aiming at closing the realism gap while preserving accurate scene alignment.

**Relighting.** Given a set of input views, scene reconstruction approaches based on radiance fields recover the outgoing radiance along with scene geometry. The radiance field depends on the scene's lighting and varies when an object is placed in a different context. In general, the outgoing radiance is governed by the rendering equation (Kajiya, 1986). An exact solution to scene relighting would involve modeling light propagation via ray tracing. Although some recent works introduce solutions for efficient ray tracing (Xie et al., 2024; Govindarajan et al., 2025; Moenne-Loccoz et al., 2024; Byrski et al., 2025), relighting remains beyond the scope of their work.

As an alternative, several recent works model light propagation using approximations to the rendering equation from real-time graphics. LumiGauss (Kaleta et al., 2024) introduces a splat-based relighting method using spherical harmonics (Ramamoorthi & Hanrahan, 2001), but it requires multi-illumination data and is restricted to diffuse surfaces. GaussianShader (Jiang et al., 2024) employs the split-sum approximation (Karis & Games, 2013) to enhance specular reflections during reconstruction. In contrast, our approach leverages a similar PBR-based shading model while enabling relighting for scenes captured under fixed illumination.

Our relighting procedure requires an environmental map. In (Liang et al., 2024), DilPIR employs a generative model to infer it using a gradient-based procedure. In turn, we estimate the environmental map using training frames with (Phongthawee et al., 2024) without additional costly optimization procedure.

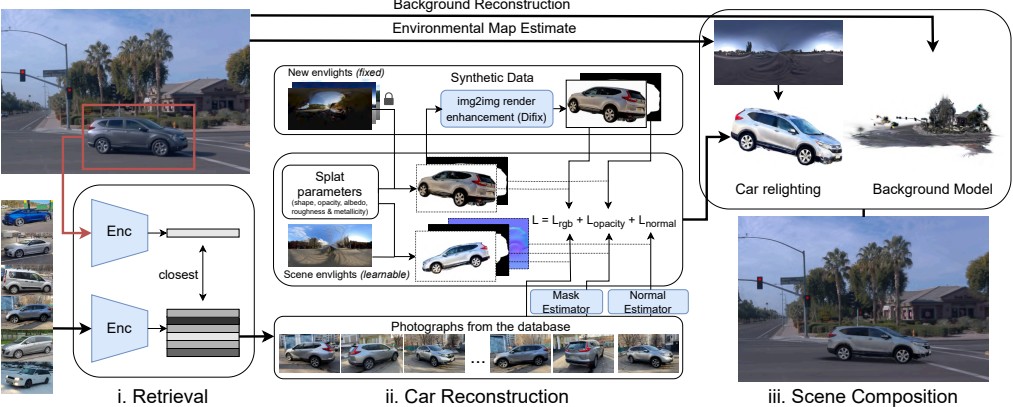

Figure 2: **MADRIVE Overview.** Given an input frame sequence, our retrieval scheme finds similar vehicles in an external database (Left). The 3D reconstruction pipeline then produces detailed vehicle models from the retrieved videos. The vehicles are represented with relightable 2D Gaussian splats. Opacity masks are used to remove background splats. The model geometry is regularized with external normals maps. (Middle). The reconstructed vehicles are adapted to the scene's lighting conditions and composed with the background to produce the overall scene representation (Right).

## 3 METHOD

In this section, we describe MADRIVE that replaces the vehicles in the scene with visually similar, fully-observed 3D car assets, thereby enabling the prediction of future vehicle appearances following sharp turns or other complex maneuvers. The overview of our method is presented in Figure 2. In the following, we describe the proposed method in detail.

### 3.1 DRIVING SCENE RECONSTRUCTION

Following (Yan et al., 2024), we decompose the scene into static and dynamic components. The static component can be reconstructed based on the video from the moving vehicle. The movement parallax and the availability of depth sensor data allow to recover the scene structure.

We adapt the approach from Street Gaussians (Khan et al., 2024) to represent the static component of the scene consisting of three parts: ground, surroundings, and sky. We parameterize the surroundings with 3D Gaussian Splats (Kerbl et al., 2023a). We represent the ground part of the scene with horizontal 2D Gaussian splats. We avoid distance estimation ambiguities by putting the sky at an infinite distance and blending it into the scene at the last step.

The dynamic component includes all moving vehicles in the scene. For simplicity, we treat cars labeled as stationary in the dataset metadata as part of the static component. In general, there are two challenges in estimating and modeling the dynamic part of the scene. First, it requires accounting for compound motion. Second, observations often capture only a limited portion of a dynamic object. For example, predicting a vehicle's side turn is difficult if its appearance from certain angles was never observed.

In line with Gaussian splatting-based urban driving scene modeling works (Yan et al., 2024; Khan et al., 2024), we represent observed vehicles as static Gaussian splats within the corresponding moving bounding box to model the compound motion in the scene. To obtain the static part of the scene, we initialize both static and dynamic parts with LIDAR data and train the splats with photometric loss.

During inference, we reuse the static part of the scene. At the same time, we replace moving vehicles with 3D car models extracted from a bank of cars using the retrieval-based approach, described in the next section. This substitution allows obtaining high-quality renders for configurations significantly diverging from the ones observed during training.

### 3.2 RETRIEVAL AND CAR RECONSTRUCTION

We propose reconstructing the dynamic part of the scene with a retrieval-based approach. Specifically, we first extract crops of the moving cars observed in the scene and find the similar car instances in the database of multi-view car captures. Then, given the retrieved images, we construct photorealistic 3D car models and replace the original cars in the scene with the obtained 3D assets. Despite

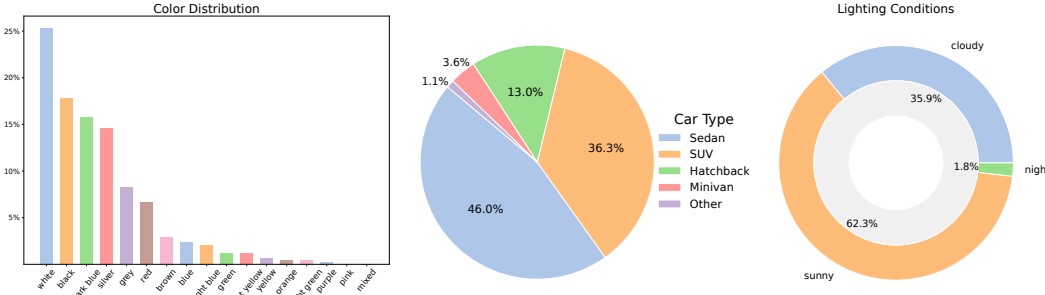

Figure 3: **MAD-Cars Analysis.** Memory-bank statistics on colors (Left), car types (Middle) and lighting conditions (Right).

the limited car visibility in the scene, retrieval-augmented reconstruction enables faithful 3D car reconstruction even from a single frame.

**Retrieval Details.** To produce a retrieval query, we compute a mask by projecting the 3D bounding box of a car onto an image plane. After filtering out small masks and overlapping masks, we use the remaining ones to extract image crops containing individual cars. For each crop, we compute an image embedding using SigLIP2 (Tschannen et al., 2025) and extract the car color using Qwen2.5-VL (Yang et al., 2024a). This color cue complements the image features, which tend to focus more on brand and car type, as observed in our experiments. To retrieve the car instance in the database, we first collect database entries with similar color and then select the one with the closest image embedding. We used YOLOv11 (Jocher & Qiu, 2024) to obtain instance segmentation masks for filtering out nearby cars in the scene. Once matches are found, we reconstruct the corresponding 3D car models using the associated multi-view image sets. The following section details our reconstruction procedure.

**Relightable Car Models.** We begin by specifying the representation used to model vehicles. By default, Gaussian splatting approximates the radiance field observed in the training frames as a whole. In our setup, however, we need to explicitly separate lighting and material effects to enable model insertion into environments with different illumination. To this end, we adopt a relighting strategy based on physically based shading (Burley & Studios, 2012).

We use a two-dimensional modification of Gaussian splats (Huang et al., 2024a), which approximates the 3D model with a collection of flat Gaussian splats. Each splat is parameterized by its location $\mu \in \mathbb{R}^3$, orientation matrix $R \in SO(3)$, transparency $\alpha \in \mathbb{R}$, and *two* scale parameters $\sigma_x, \sigma_y \in \mathbb{R}$. Unlike 3D splats, 2D splats have well-defined surface normals $\mathbf{n} = \mathbf{n}(R)$, which are essential for surface relighting effects.

To disentangle scene lighting from surface materials, we adopt the lighting model from (Munkberg et al., 2022) for each splat. The model assumes distant illumination with incident radiance $L_i(\omega_i)$ and defines the outgoing radiance in direction $\omega_o$ according to the rendering equation (Kajiya, 1986):

$$L(\omega_o) = \int_\Omega L_i(\omega_i) f(\omega_i, \omega_o)(\omega_i \cdot \mathbf{n}) \mathrm{d}\omega_i, \tag{1}$$

where $f(\omega_i, \omega_o)$ is the surface BSDF and the integration is taken over the hemisphere $\Omega$ around the surface point. The environment lighting $L_i$ is parameterized as a high-resolution cubemap. Following (Munkberg et al., 2022), we parameterize each splat's BSDF using the Cook–Torrance shading model (Cook & Torrance, 1982), with appearance defined by albedo $c \in \mathbb{R}^3$, roughness $r \in \mathbb{R}$, and metallicity $m \in \mathbb{R}$.

Finally, to avoid the cost of directly evaluating Eq. 1, we employ the differentiable split-sum approximation from (Munkberg et al., 2022), which allows us to jointly infer incident radiance and splat BSDF parameters during optimization.

**Car Reconstruction Details.** Next, we specify the details of the reconstruction algorithm used for the representation above.

For a rendered frame $I_i$ and the ground truth frame $\hat{I}_i$, our objective consists of image-based loss $\mathcal{L}_{\text{rgb}} = \mathcal{L}_1(I_i, \hat{I}_i) + \mathcal{L}_{SSIM}(I_i, \hat{I}_i)$ along with several regularizers. To exclude unnecessary background objects from the model, we generate masks $\hat{M}_i(x, y) = [\hat{I}_i(x, y) \text{ } \textit{is part of a car}]$ with

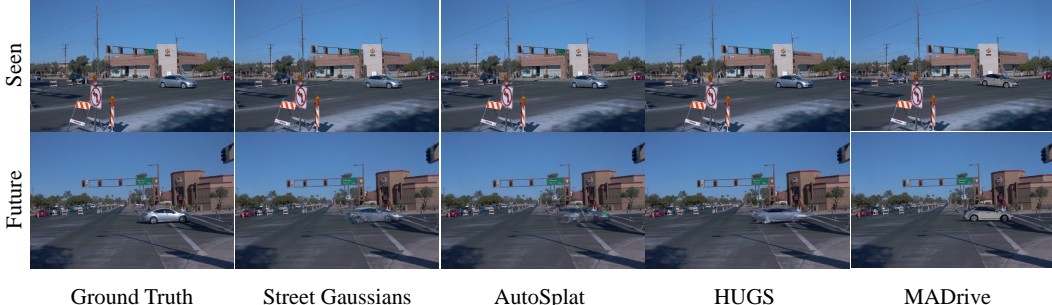

Figure 4: **Qualitative comparison** of MADRIVE with non-retrieval-based driving scene reconstruction methods. Reconstruction of the training views (Top). Reconstruction of the hold-out (future) views (Bottom).

Mask2Former (Cheng et al., 2022) to indicate pixels that belong to the model. Our opacity loss promotes high transparency outside of car pixels $\mathcal{L}_{opacity} = \sum_{x,y}(1 - \hat{M}_i) \cdot T_i$, where $T_i$ is the transparency map of the rendered frame. In our model, proper relighting requires accurate surface normals, so we additionally estimate normal maps $\hat{N}_i = n(\hat{I}_i)$ with a NormalCrafter model (Bin et al., 2025) and use the estimates to regularize Gaussian orientations. For the rendered normal maps $N_i$, the regularizer is $\mathcal{L}_{normal} = \sum_{x,y} \hat{M}_i \cdot (1 - N_i^T \hat{N}_i)$. The resulting objective is

$$\mathcal{L}_{gt}(I_i, \hat{I}_i) = \mathcal{L}_{rgb}(I_i, \hat{I}_i) + \lambda_{opacity}\mathcal{L}_{opacity}(I_i, \hat{I}_i) + \lambda_{normal}\mathcal{L}_{normal}(I_i, \hat{I}_i). \tag{2}$$

Furthermore, to promote realistic appearance under novel lighting conditions, we incorporate additional synthetic data. Disentangling illumination and scene materials is particularly challenging for in-the-wild captures, where lighting remains fixed throughout training. To address this limitation, we approximate a multi-illumination setup using synthetic data. Specifically, we render random model views under varying pre-defined environmental lighting and enhance these renders using the Difix image-to-image model (Wu et al., 2025a). The enhanced images $\tilde{I}_i$ are then used as training samples. To compute the objective in Eq. 2, we render frames $I_i$ with the same environmental lighting as in $\tilde{I}_i$, use the $\alpha$-channel of $\tilde{I}_i$ as the mask, and omit the normal regularization. We add the synthetic frames in the same proportion as the real frames. Synthetic frames are introduced after the initial 10k gradient steps and regenerated every 2.5k steps throughout the following 20k steps.

### 3.3 CAR INSERTION AND RELIGHTING

The final stage of our pipeline integrates reconstructed cars into the learned scene. First, we prepare each car for the insertion. We remove occasional splats that are either positioned behind the training cameras or project onto pixels outside the car mask in the training images. The car is then oriented based on the principal components of its point cloud, further refined using an orientation model (Scarvelis et al., 2024) to ensure proper alignment within the driving scene.

Next, the aligned point cloud is placed inside the bounding box of the original car to be replaced. To achieve precise alignment in both scale and position, we use the Iterative Closest Point (ICP) algorithm (Besl & McKay, 1992) and apply the resulting transformation to the inserted car. To enhance visual realism, we add a shadow beneath the car, modeled as a black plane composed of 2D splats placed under the wheels. While more sophisticated shadow placement based on sun position, as explored in (Zhou et al., 2024a), could be considered, we find it non-essential for our method.

Since the retrieved car asset is captured under different lighting conditions, we estimate the target scene's environment map and adjust the car's appearance via Eq. 1 to ensure visual consistency. As the Waymo dataset (Sun et al., 2020) lacks full 360° camera coverage, we approximate lighting conditions using DiffusionLight (Phongthawee et al., 2024), which reconstructs missing environment map regions via diffusion-based inpainting. Given a training frame, we estimate the environment map and align it with the corresponding camera orientation. In addition, we adjust the scale of the environmental map to minimize the tone discrepancy between the last training frame and our render.

### 3.4 DATABASE COLLECTION AND STATISTICS

This work introduces MAD-CARS, a large-scale database of multi-view car videos in the wild, sourced from online car sale advertisements. The database contains $\sim 70,000$ diverse video instances, each averaging $\sim 85$ frames, with most car instances available at a resolution of $1920 \times 1080$.

It includes cars from ∼150 brands, covering a broad range of colors, car types, and three lighting conditions. Distributions of color, car type, and lighting are illustrated in Figure 3. The metadata for each car instance is presented in the dataset.

The data is carefully curated by filtering out frames and entire car instances that could negatively impact 3D reconstruction. In more detail, we remove low quality and overly dark frames with CLIP-IQA (Wang et al., 2023a) and use Qwen2.5-VL (Yang et al., 2024a) to detect finger blocked shots, car interior views, the frames where the car view is occluded, e.g., by fences, trees, other vehicles, etc. More data collection details are provided in Appendix A.

## 4 EXPERIMENTS

In the following, we report the MADRIVE performance. Section 4.1 describes our evaluation setup. Then, we proceed to the main results in Section 4.2. Finally, we explore the retrieval, car reconstruction and relighting procedures in Section Section 4.3.

### 4.1 EVALUATION SETUP

**Scene Reconstruction Dataset.** We reconstruct the driving scenes from the Waymo Open Motion dataset (Ettinger et al., 2021). We picked 12 particularly challenging scenes containing multiple cars, driving maneuvers and diverse lighting conditions. Then, we manually select scene sequences and divide them into training and evaluation clips. In our experiments, we simultaneously use videos from frontal and two side cameras to capture a wide field of view and track cars moving across the scene. More evaluation setup details are provided in Appendix B.

**Scene Extrapolation with Novel View Synthesis.** For our evaluation, we selected driving scenes involving U-turns, intersection crossings, and parking departures — common accident scenarios that also reveal vehicles from diverse viewpoints, posing challenges for reconstruction. Each frame sequence was manually split into training and testing subsets at the midpoint of the maneuver. We use the whole sequence to reconstruct the background and then remove the cars using the annotated bounding boxes in the Waymo dataset. Car reconstruction is performed using only the first part of the sequence, while the second part is reserved for evaluating scene reconstruction quality. We aim to generate realistic views of the scene by extrapolating the observed data. In particular, we insert the reconstructed car models into the background according to location and orientation specified by the bounding boxes on the holdout sequence. By design, our setup evaluates scenes under configurations that differ significantly from the frames seen during training. At the same time, the data split ensures that test frames do not leak into the car reconstruction process.

**Baselines.** We compare MADRIVE with the scene reconstruction Gaussian splatting-based methods that were previously considered for novel view synthesis: Street-Gaussians (SG) (Yan et al., 2024), AutoSplat (Khan et al., 2024) (our implementation), and HUGS (Zhou et al., 2024b). Details on training and evaluation of baselines are given in Appendix D.

### 4.2 MAIN EXPERIMENTS

**Qualitative Evaluation.** First, we provide visual scene reconstruction results for qualitative analysis. In Figure 4, we compare rendering results on the training and hold-out frames. Although SG, AutoSplat, and HUGS produce accurate approximations of training frames, on the test frames cars tend to fell apart for novel view angles. Compared to baselines, our method cannot reproduce the training frames with the same precision, but is significantly more robust to deviations from training configurations. More visual examples are presented in Figures 9, 10. We also provide the visualizations with modified trajectories in Figure 11.

Table 1: Comparison in terms of tracking and segmentation metrics.

| Model | MOTA ↑ | MOTP ↓ | IDF1 ↑ | Segmentation IoU ↑ |
|---|---|---|---|---|
| Street-GS (Yan et al., 2024) | 0.654 | **0.105** | 0.776 | 0.556 |
| HUGS (Zhou et al., 2024b) | 0.556 | 0.221 | 0.699 | 0.333 |
| AutoSplat[*] (Khan et al., 2024) | 0.589 | 0.154 | 0.716 | 0.489 |
| MADRIVE (Ours) | **0.841** | 0.138 | **0.913** | **0.818** |

[*]Denotes our reimplementation.

**Quantitative Evaluation.** In our main experiments, we assess tracking and segmentation performance on synthesized **test** frames. Specifically, we apply state-of-the-art tracking and segmentation

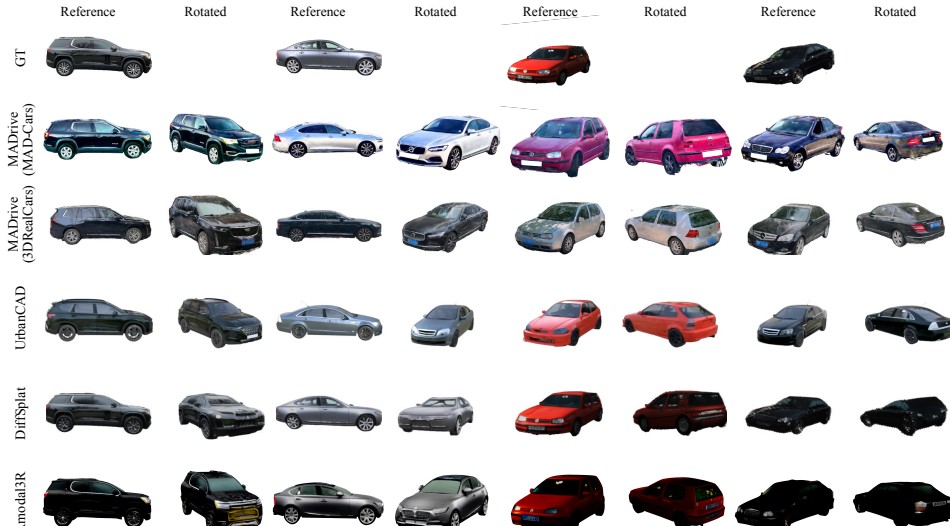

Figure 5: **Qualitative comparison of reconstructed vehicles** on KITTI-360 from the reference and rotated viewpoints. MADRIVE on top of the MAD-CARS dataset produces more similar and realistic 3D assets.

models to both synthesized and ground truth frames and compare their outputs using established metrics for each task. For tracking, we use BotSort (Aharon et al., 2022) with a YOLOv8n backbone, reporting multiple object tracking accuracy (MOTA↑), precision (MOTP↓), and identity F1 score (IDF1) (Milan et al., 2016). For segmentation, we compute the average intersection-over-union (IoU) using instance segmentation masks obtained with Mask2Former (Cheng et al., 2022). Table 1 presents the comparison of MADRIVE against the baselines. MADRIVE shows substantially superior performance compared to the baselines in 2 tracking metrics (MOTA and IDF) and segmentation metric IoU. This observation is also supported by the visual examples provided in Figure 4. We explain the MOTP gap between our method and Street-GS by the better car alignment of Street-GS in the first test frames, while later frames, where the tracker fails to detect Street-GS cars, are not counted in the MOTP calculation. We provide per scene results for all 12 scenes in Appendix C. We also discuss the choice for the reference masks in the evaluation protocol in Appendix E.

## 4.3 FURTHER EVALUATION

**Retrieval.** Here, we evaluate the performance of the proposed retrieval module isolated from other components to address how accurately the retrieved cars correspond to the original cars in the scene.

We compare the retrieval performance on the proposed dataset against 3DRealCars (Du et al., 2024), highly accurate publicly available dataset of 2,500 car assets. To evaluate the retrieval quality, we first calculate the average L2 distance between the car images from the driving scene and the nearest cars from the memory bank. We use SigLIP2 So (Tschannen et al., 2025) as an image feature extractor. Then, we provide the accuracy obtained with

Table 2: Retrieval performance w/o color filtering in terms of accuracy on the car brand, model, color and type and the distance to the closest instance for the MAD-CARS and 3DRealCar (Du et al., 2024) datasets. MAD-CARS enables more accurate retrieval of cars across all attributes.

| Dataset | Brand ↑ | Model ↑ | Color ↑ | Car Type ↑ | Distance ↓ |
|---|---|---|---|---|---|
| 3DRealCars | 0.626 | 0.503 | 0.508 | 0.888 | 0.502 |
| MAD-CARS | 0.750 | 0.663 | 0.533 | 0.913 | 0.445 |

the Qwen2.5-VL-32B-Instruct model, asked to compare the cars in terms of their brand, model, color and car type. For a fair comparison, we do not use the color filtering in this experiment.

Table 2 shows the retrieval accuracy across different attributes, and the average L2 distance to the closest instance. We observe that candidates retrieved using MAD-CARS more accurately match the cars in the driving scenes, which we attribute to the significantly larger scale of MAD-CARS.

Importantly, Table 2 highlights that retrieval based solely on feature extractors often disregards car color, despite its importance for realistic car replacement. The similar problem has been observed for non-visual-language encoders such as DINOv2 (Oquab et al., 2023). Our additional results

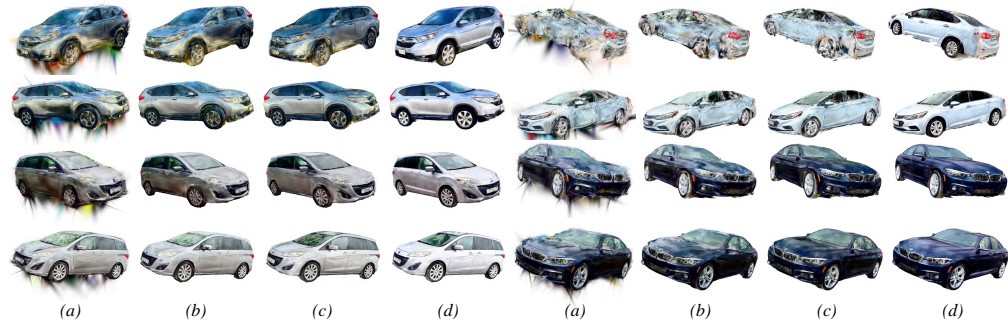

| | | | | | | | |
|---|---|---|---|---|---|---|---|
| *(a)* | *(b)* | *(c)* | *(d)* | *(a)* | *(b)* | *(c)* | *(d)* |

Figure 6: Ablation of reconstruction regularizers, each setting adds one component to the previous and shows albedo. (a) Without regularization, shape and texture artifacts cause uneven edges. (b) Adding opacity regularization improves edge quality. (c) Adding normal regularization enhances surface smoothness. (d) Training with synthetic frames under varying lighting disentangles illumination from object color, yielding cleaner albedo and reconstructions.

in Appendix F show that applying a color-based pre-filtering improves color consistency between the retrieved and target vehicles.

**Car reconstruction.** We provide a qualitative comparison with other car reconstruction approaches in Figure 5, where we visualized reconstruction alternatives. Given a query frame from the KITTI dataset, we compared the proposed approach with three alternatives: car reconstruction on a different car dataset (Du et al., 2024), matching with a car model from a CAD dataset (Lu et al., 2024), and running a cutting edge image-to-3D models (Lin et al., 2025; Wu et al., 2025b). Even though the latter closely matches the query frame, the second view indicates a subpar geometry recovery. Compared to other methods, we see that the diversity of our dataset allows MADRIVE to obtain models that closely match the query frame in terms of appearance (e.g., color, shape) and realism.

We further ablate the components of the proposed reconstruction algorithm. Figure 6 shows the recovered geometry and albedo for several cars. Reconstruction without regularization (a) produces noticeable artifacts in both shape and texture, leading to uneven edges after background removal. Introducing opacity regularization to suppress the background during reconstruction (b) improves edge quality. Adding normal regularization (c) further enhances surface smoothness and consistency. Finally, incorporating synthetic frames under varying lighting conditions (d) helps disentangle scene illumination from object color, yielding more accurate albedo and overall cleaner reconstructions.

**Relighting.** We conclude with a qualitative comparison of the proposed relighting scheme. For a number of scenes, we reconstructed scene frames with and without the relighting module. Figure 7 shows that the relighting module adapts the model colors to the environment, helping to reduce the inconsistencies that break immersion and make the inserted models appear naturally lit within the scene.

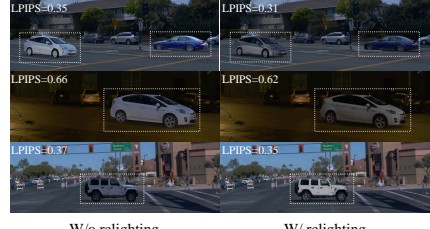

Figure 7: **Relighting ablation.** Rendered hold-out frames without (Left) and with (Right) relighting.

## 5 CONCLUSION

This work presents MADRIVE, a novel driving scene reconstruction approach specifically designed to model significantly altered vehicle positions. Powered by MAD-CARS, our large-scale multi-view car dataset, MADRIVE replaces dynamic vehicles in a scene with similar car instances from the database. We believe that MADRIVE could make a step towards modeling multiple potential outcomes for analyzing an autonomous driving system's behavior in safety-critical situations.

However, despite the promising visual fidelity of future scenario frames, they still differ from the ground truth, as we discuss in Appendix I. Future work could focus on expanding the database with a wider range of car brands, models, and types, as well as enhancing corrupted car videos using recent multi-view diffusion (Zhou et al., 2025).

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

## A   DATA COLLECTION DETAILS

The initial database contained $\sim 95,000$ car videos of $\sim 100$ views on average. The first filtering stage includes the filtering of low quality and overly dark images with the CLIP-IQA model (Wang et al., 2023a), discarding frames with a score $< 0.2$. Then, we use Qwen-2.5-VL-Instruct (7B) (Yang et al., 2024a) to respond several questions for each frame:

- "Does the image depict a car?"
- "Is the car directly occluded?"
- "Does the image depict the car interior?"
- "Does a hand or finger block the view?"
- "Is the car door open?"
- "Does the image mainly depict the car window?"

Based on the responses, we filter out the corresponding frames or, in some cases, entire car instances. Also, if fewer than 45 valid frames remain for a given instance, the entire instance is discarded.

## B   EVALUATION SETUP DETAILS

For scene reconstruction evaluation, we selected 12 scenes from the Waymo Open Dataset (Sun et al., 2020), with labels listed in Table 3. This table also provides the correspondence between the original scene labels from the Waymo Cloud Storage and the short names used in our work. We split each scene into training and testing subsets based on time (Table 4) and camera selection (Table 5). Specifically, frames with indices $i^{\text{train}}$, where $i^{\text{train}} \in [i^{\text{train}}_{\text{start}}, i^{\text{train}}_{\text{end}}]$, were used for training. For evaluation, we used frames $i^{\text{test}} \in [i^{\text{test}}_{\text{start}}, i^{\text{test}}_{\text{end}}]$, with all split indices provided in Table 4.

Table 3: Waymo scenes used for evaluation of scene reconstruction.

| Label | Scene name |
|---|---|
| 1231623110026745648_480_000_500_000 | 123 |
| 1432918953215186312_5101_320_5121_320 | 143 |
| 1906113358876584689_1359_560_1379_560 | 190 |
| 10500357041547037089_1474_800_1494_800 | 105 |
| 10940952441434390507_1888_710_1908_710 | 109 |
| 16504318334867223853_480_000_500_000 | 165 |
| 17407069523496279950_4354_900_4374_900 | 174 |
| 18025338595059503802_571_216_591_216 | 180 |
| 14183710428479823719_3140_000_3160_000 | 141 |
| 15834329472172048691_2956_760_2976_760 | 158 |
| 17647858901077503501_1500_000_1520_000 | 176 |
| 7799671367768576481_260_000_280_000 | 779 |

## C   PER-SCENE QUANTITATIVE EVALUATION.

In addition to the aggregated results in Table 1, we report per-scene metric values in Table 6, Table 7, Table 8, and Table 9, corresponding to MOTA, MOTP, IDF1, and IoU, respectively. We observe that MADRIVE consistently outperforms the baselines across most scenes.

## D   BASELINE DETAILS

**Baselines training and evaluation.** We trained all baselines (Street-Gaussians, HUGS, and AutoSplat) for $10K$ iterations using the training frames with indices $i \in [i^{\text{train}}_{\text{start}}, i^{\text{train}}_{\text{end}}]$ as specified in Table 4. Additionally, we trained the background models for both the baselines and MADRIVE for $30K$ iterations using all available frames. These pretrained background models were then used during the rendering of the test frames ($i \in [i^{\text{test}}_{\text{start}}, i^{\text{test}}_{\text{end}}]$), on which we compute the metrics reported in Table 6, Table 7, Table 8, and Table 9.

Table 4: Train and test frame splits for Waymo scenes over time. All values, except those in the leftmost column, indicate frame indices starting from 0.

| Scene name | $i_{start}^{train}$ | $i_{end}^{train}$ | $i_{start}^{test}$ | $i_{end}^{test}$ |
|---|---|---|---|---|
| 123 | 106 | 116 | 117 | 175 |
| 143 | 43 | 53 | 54 | 62 |
| 190 | 115 | 125 | 126 | 137 |
| 105 | 164 | 174 | 175 | 196 |
| 109 | 1 | 16 | 17 | 55 |
| 165 | 7 | 40 | 41 | 111 |
| 174 | 34 | 51 | 52 | 72 |
| 180 | 49 | 55 | 56 | 68 |
| 141 | 60 | 80 | 81 | 117 |
| 158 | 44 | 62 | 63 | 100 |
| 176 | 31 | 42 | 43 | 67 |
| 779 | 50 | 65 | 66 | 84 |

Table 5: Train and test frame splits for Waymo scenes based on camera selection.

| Scene name | Train cameras | Test cameras |
|---|---|---|
| 123 | frontal, frontal left | frontal, frontal left |
| 143 | frontal, frontal left | frontal, frontal left |
| 190 | frontal, frontal left | frontal, frontal left |
| 105 | frontal, frontal left | frontal |
| 109 | frontal, frontal right | frontal right |
| 165 | frontal, frontal left | frontal, frontal left |
| 174 | frontal | frontal |
| 180 | frontal, frontal right | frontal, frontal right |
| 141 | frontal | frontal |
| 158 | frontal | frontal |
| 176 | frontal | frontal |
| 779 | frontal, frontal left, frontal right | frontal, frontal right |

**Street-Gaussians.** We used the official implementation available at `https://github.com/zju3dv/street_gaussians`.

**HUGS.** We used the official implementation provided at `https://github.com/hyzhou404/HUGSIM`.

**AutoSplat.** As no official implementation is publicly available, we re-implemented the core contributions of AutoSplat on top of the Street-Gaussians codebase.

# E CHOICE OF REFERENCE MASKS IN THE EVALUATION

In our validation setup, we used predictions from tracking and segmentation models on ground-truth images as targets, since the Waymo dataset lacks segmentation masks and 2D bounding boxes.

To evaluate whether cars in the synthesized frames are as identifiable as those in the original frames, we applied the same detection algorithm to both. Our method outperforms the baseline, primarily because our system inserts visually coherent cars on test frames by leveraging reconstructed models, whereas baseline approaches result in degraded or incomplete vehicle representations.

However, the inserted cars might be easier to detect. To test this, we conducted an additional experiment. Specifically, we generated a new set of detector targets by projecting the 3D bounding boxes provided in the Waymo dataset onto the image plane. We then evaluated the performance of the detector on both the ground-truth and synthesized (MADrive) frames using the new "ground-truth" annotation.

Table 6: Mean MOTA ↑ results on test frames for all Waymo scenes.

| Scene name | SG | HUGS | AutoSplat | MADRIVE |
|---|---|---|---|---|
| 123 | 0.687 | 0.685 | 0.327 | 0.887 |
| 143 | 0.650 | 0.513 | 0.600 | 0.825 |
| 190 | 0.787 | 0.795 | 0.904 | 0.858 |
| 105 | 0.906 | 0.656 | 0.742 | 0.906 |
| 109 | 0.242 | 0.448 | 0.605 | 0.925 |
| 165 | 0.684 | 0.461 | 0.788 | 0.883 |
| 174 | 0.809 | 0.886 | 0.830 | 0.936 |
| 180 | 0.611 | 0.528 | 0.695 | 0.778 |
| 141 | 0.667 | 0.607 | 0.163 | 0.767 |
| 158 | 0.423 | 0.233 | 0.681 | 0.639 |
| 176 | 0.727 | 0.562 | 0.176 | 0.912 |
| 779 | 0.661 | 0.296 | 0.545 | 0.779 |

Table 7: Mean MOTP ↓ results on test frames for all Waymo scenes.

| Scene name | SG | HUGS | AutoSplat | MADRIVE |
|---|---|---|---|---|
| 123 | 0.073 | 0.093 | 0.099 | 0.079 |
| 143 | 0.114 | 0.461 | 0.095 | 0.203 |
| 190 | 0.088 | 0.112 | 0.115 | 0.144 |
| 105 | 0.073 | 0.262 | 0.222 | 0.118 |
| 109 | 0.093 | 0.132 | 0.094 | 0.122 |
| 165 | 0.125 | 0.202 | 0.119 | 0.149 |
| 174 | 0.075 | 0.886 | 0.078 | 0.093 |
| 180 | 0.150 | 0.231 | 0.194 | 0.195 |
| 141 | 0.119 | 0.261 | 0.237 | 0.179 |
| 158 | 0.087 | 0.128 | 0.123 | 0.119 |
| 176 | 0.093 | 0.246 | 0.167 | 0.072 |
| 779 | 0.176 | 0.443 | 0.305 | 0.180 |

The results in Table 10 show that the predictions on ground-truth images align slightly better with the projected 3D bounding boxes than those on the synthesized MADrive frames. This indicates that our inserted cars do not artificially simplify detection, supporting the validity of our evaluation.

## F    ADDITIONAL RETRIEVAL EVALUATION RESULTS

In this section, we provide an additional illustration of our retrieval algorithm. As shown in our Figure 8, introducing a color-based pre-filter enhances the alignment of vehicle colors between retrieved candidates and the target.

## G    EVALUATION OF INSERTED MODEL QUALITY

As mentioned earlier, an alternative to our approach would be to generate car models using a pre-trained image-to-3D generative model. However, such methods typically produce low-resolution cars with limited detail and a cartoon-like appearance. To further assess the viability of this alternative, we compared renderings from both approaches against a hold-out set of real car images from MAD-CARS. Specifically, given car crops from driving scenes, we reconstructed models using our retrieval-based approach and a state-of-the-art image-to-3D method, Amodal3R (Wu et al., 2025b). To reduce domain shift, backgrounds were excluded from MAD-CARS. The results in Table 11 support our claim that the retrieval-augmented approach yields cars with higher resemblance to real ones.

Table 8: Mean IDF1 ↑ results on test frames for all Waymo scenes.

| Scene name | SG | HUGS | AutoSplat | MADRIVE |
|---|---|---|---|---|
| 123 | 0.804 | 0.806 | 0.475 | 0.940 |
| 143 | 0.787 | 0.709 | 0.750 | 0.904 |
| 190 | 0.880 | 0.887 | 0.950 | 0.924 |
| 105 | 0.952 | 0.780 | 0.877 | 0.951 |
| 109 | 0.390 | 0.619 | 0.754 | 0.961 |
| 165 | 0.806 | 0.612 | 0.894 | 0.936 |
| 174 | 0.894 | 0.940 | 0.907 | 0.967 |
| 180 | 0.753 | 0.709 | 0.820 | 0.871 |
| 141 | 0.805 | 0.698 | 0.278 | 0.866 |
| 158 | 0.605 | 0.377 | 0.829 | 0.797 |
| 176 | 0.847 | 0.720 | 0.316 | 0.955 |
| 779 | 0.793 | 0.532 | 0.739 | 0.881 |

Table 9: Mean IoU ↑ results on test frames for all Waymo scenes.

| Scene name | SG | HUGS | AutoSplat | MADRIVE |
|---|---|---|---|---|
| 123 | 0.753 | 0.608 | 0.500 | 0.866 |
| 143 | 0.485 | 0.243 | 0.510 | 0.779 |
| 190 | 0.707 | 0.519 | 0.740 | 0.846 |
| 105 | 0.671 | 0.425 | 0.439 | 0.731 |
| 109 | 0.499 | 0.246 | 0.419 | 0.832 |
| 165 | 0.633 | 0.459 | 0.647 | 0.730 |
| 174 | 0.695 | 0.581 | 0.655 | 0.829 |
| 180 | 0.475 | 0.238 | 0.582 | 0.814 |
| 141 | 0.607 | 0.196 | 0.226 | 0.765 |
| 158 | 0.404 | 0.135 | 0.498 | 0.862 |
| 176 | 0.499 | 0.187 | 0.263 | 0.886 |
| 779 | 0.247 | 0.153 | 0.273 | 0.874 |

# H  ADDITIONAL QUALITATIVE COMPARISONS AND NEW TRAJECTORIES

We provide additional visual results in Figures 9 and 10. We also demonstrate our method's capability to render novel views with substantial scene variations. Figure 11 showcases results across four test scenes, where all modifications preserve high image quality.

# I  LIMITATIONS

**Trade-off on seen and unseen data.**  Figures 4, 5, 9 , 10 show that baseline methods achieve high photometric consistency by optimizing on training frames, while MADRIVE have visual differences.

However, baseline methods significantly degrade on unseen test frames, which is critical for our initial goal. This gap highlights a trade-off of our design: although the visual fidelity is lower at training, our method enables fast and scalable simulation of diverse unseen scenarios.

Importantly, our system is fully automated and requires no human intervention: the retrieval, placement, and orientation of car models are all handled automatically. Our validation experiments demonstrate that the generated scenes allow vehicle perception modules to reasonably assess the depicted traffic situations.

**Reconstruction limitations.**  To run reconstruction, we estimate camera parameters from the input images. In particular, we run bundle adjustment starting from the initialization obtainev with VGGT. At present, errors in camera estimation remain a primary source of reconstruction failures. We expect that continued advances in foundational vision models will substantially reduce this limitation.

State-of-the-art multiview reconstruction methods continue to struggle with reflective and glossy surfaces like cars even up to this day. Accurate modeling of reflections on metallic surfaces on real

Table 10: Comparison between detector performance on both the ground-truth and synthesized (MADrive) frames using projected 3D bounding boxes.

| Model | MOTA ↑ | MOTP ↓ | IDF1 ↑ |
|---|---|---|---|
| GT frames | **0.879** | **0.270** | **0.928** |
| MADRIVE frames | 0.861 | 0.340 | 0.908 |

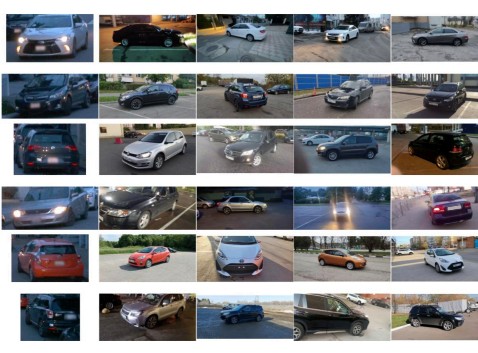 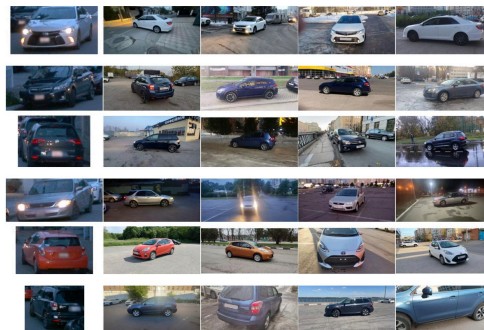

Query    Top-4 candidates, w/o color filtering    Query    Top-4 candidates, w/ color filtering

Figure 8: **Retrieval illustration.** Top-4 candidates retrieved using SigLIP 2 without (Left) and with (Right) color filtering.

datasets demands more precise representations of illumination - beyond what conventional environment maps can provide.

## J   STATEMENT ON LLM USAGE

The authors used the large language model (LLM) only to improve the writing and grammar of the text. All the results from the LLM were checked by the authors.

Table 11: Quantitative assessment of car model quality, comparing image-to-3D generative models with a retrieval-augmented approach.

| | FID ↓ | KID×$10^3$ ↓ |
|---|---|---|
| Amodal3R | 81.65 | 51.91 |
| MADrive frames | 62.64 | 39.40 |

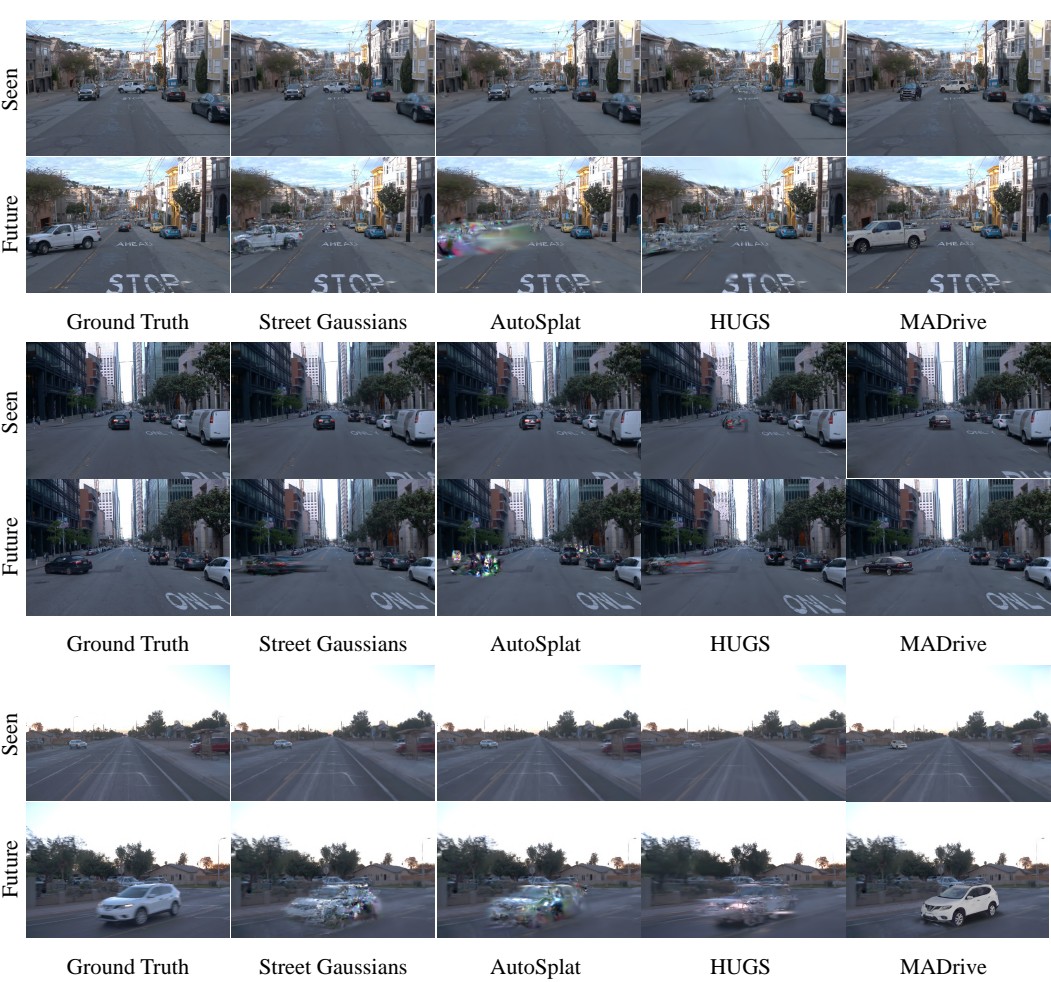

Figure 9: **Additional qualitative comparison** of MADRIVE with non-retrieval-based driving scene reconstruction methods. Reconstruction of the training views (Top). Reconstruction of the hold-out (future) views (Bottom).

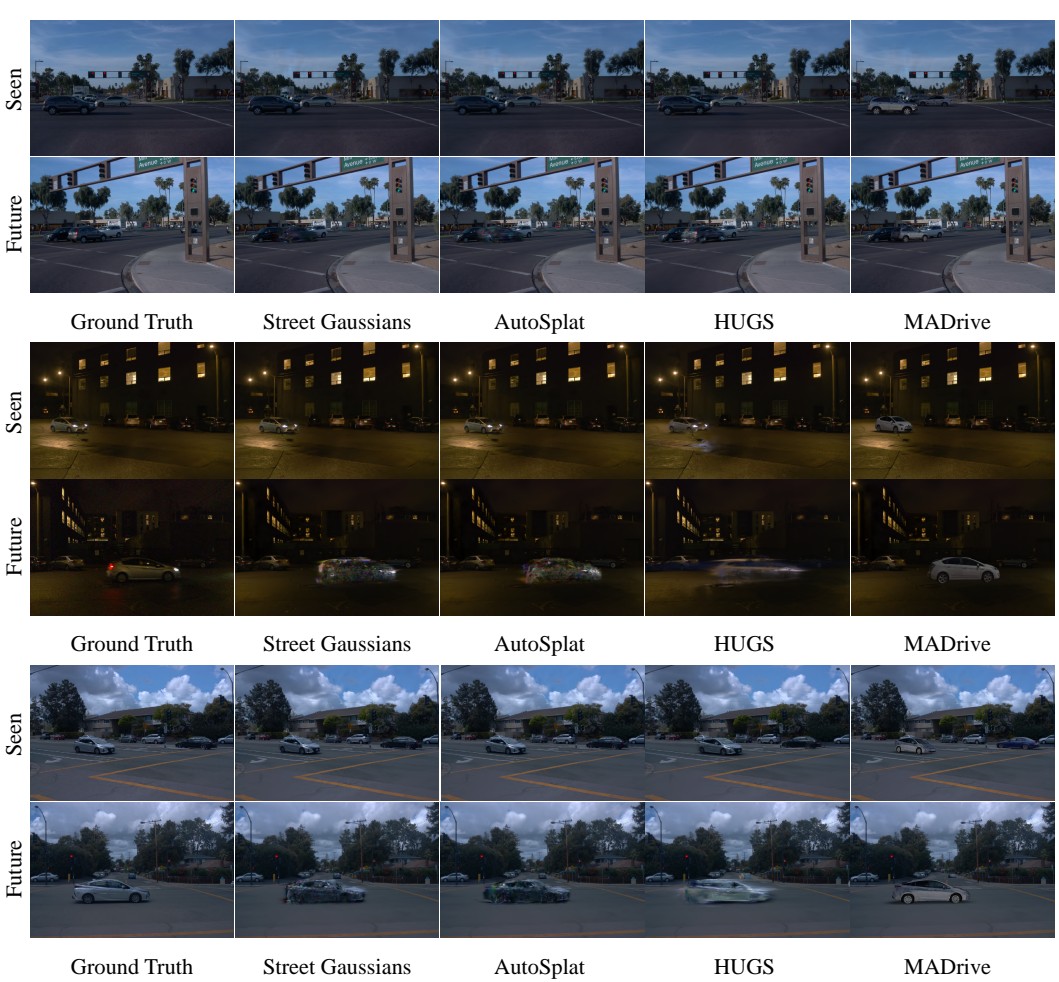

Figure 10: **Additional qualitative comparison** of MADRIVE with non-retrieval-based driving scene reconstruction methods. Reconstruction of the training views (Top). Reconstruction of the hold-out (future) views (Bottom).

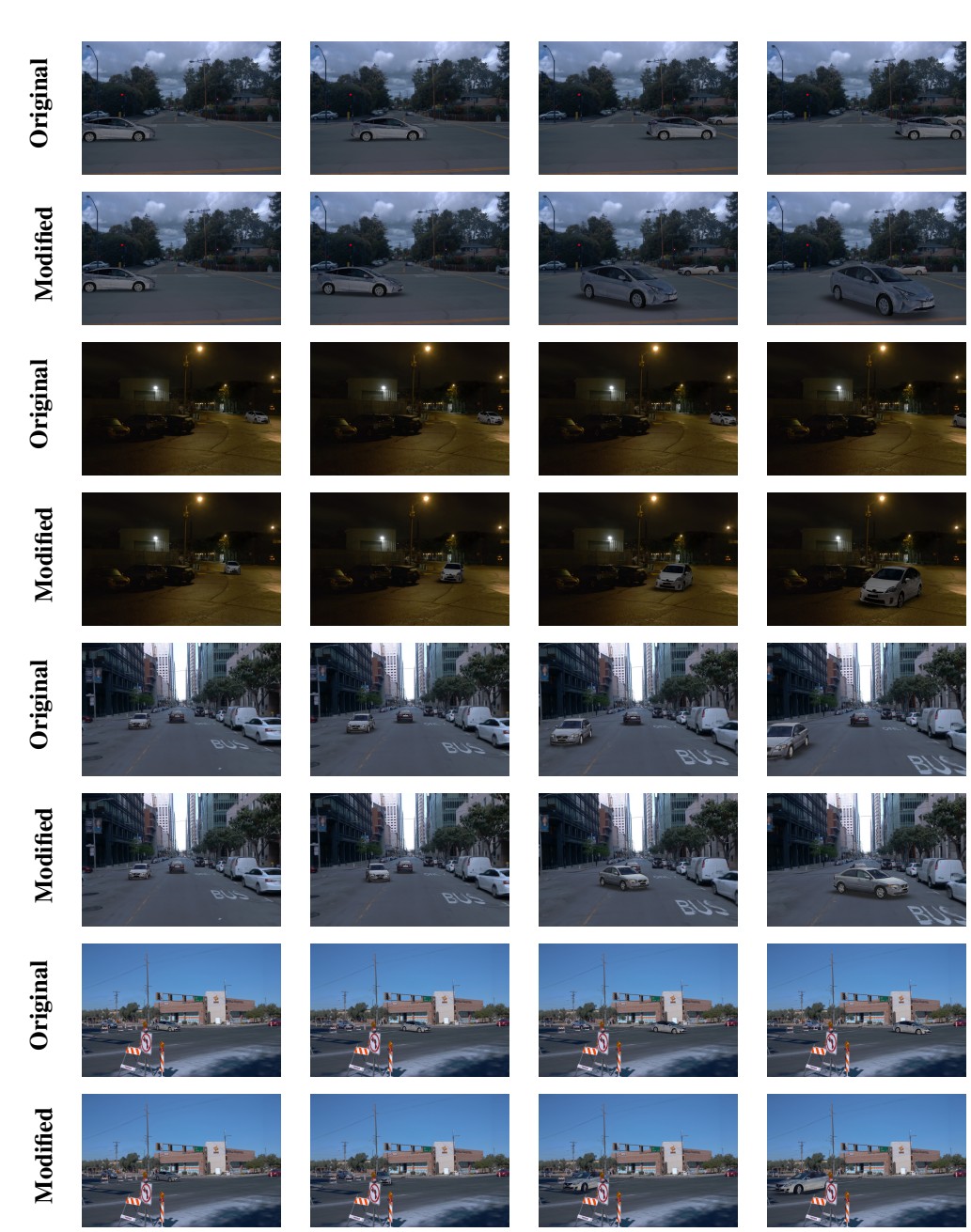

Figure 11: Visualization of original and modified trajectories with MADRIVE. The cars retain high-fidelity appearance even at close distances to the ego camera.

