# OpenReview forum: "MADrive: Memory-Augmented Driving Scene Modeling"
_ICLR.cc/2026/Conference — ICLR 2026 Conference Withdrawn Submission_

### Official Review · Reviewer_SnEq · 2025-10-23

**Soundness:** 3
**Presentation:** 3
**Contribution:** 3
**Rating:** 4
**Confidence:** 5

**Summary:**

Recent advances in scene reconstruction leverage 3D Gaussian splatting for highly realistic autonomous driving (AD) environment modeling. However, such reconstructions are tightly linked to original observations and fail to support photorealistic synthesis of heavily altered or novel driving scenarios. This work proposes MADrive—a memory-augmented reconstruction framework that extends existing methods by replacing observed vehicles with visually similar 3D assets from a large external memory bank. Specifically, we release MAD-Cars, a curated dataset of K 360° in-the-wild car videos, and a retrieval module: it identifies the most similar car instances in the memory bank, reconstructs their 3D assets from video, and integrates them into the target scene via orientation alignment and relighting. Experiments show these replacements enable complete multi-view vehicle representations, supporting photorealistic synthesis of significantly altered configurations.

**Strengths:**

Clear motives and interesting research questions​

Clear method design and comprehensive technical details​

Detailed and rigorous experimental design​

**Weaknesses:**

1. The image quality in the paper is subpar, making them unsuitable for practical scenario applications. Clear, high-resolution images are essential for effectively illustrating key findings and ensuring the reproducibility of results, which the current figures fail to achieve.

2. The reconstruction results presented in the demo appear to be of low quality. Specific issues (e.g., blurred details, inaccurate structural restoration, or inconsistent texture mapping) are not adequately addressed, and the demo does not effectively demonstrate the method’s performance in solving the targeted problem.

3. There seem to be multiple alternative solutions for addressing the problem proposed in the paper, and the current approach does not stand out as the optimal one. For instance, the method introduced in *Editable Scene Simulation for Autonomous Driving via LLM-Agent Collaboration* also targets a similar problem domain and may offer more comprehensive or efficient solutions. The paper lacks a thorough comparative analysis to justify why its proposed approach is superior to such existing alternatives.

4. The methodological innovation of this work is limited. The core ideas, technical routes, or experimental designs do not present significant breakthroughs compared to state-of-the-art studies in the field. Most components of the method either build on existing frameworks without substantial improvements or combine conventional techniques in a non-novel manner.

5. Could you provide images of vehicles captured from different viewing angles? Supplementary multi-view vehicle images would help verify the method’s robustness across varying perspectives, which is particularly critical if the work involves tasks such as vehicle detection, reconstruction, or scene simulation related to multi-angle scenarios.

**Questions:**

The drawbacks are listed above. My main consideration is his practicality and effectiveness. It has a significant gap compared to the results of many existing algorithms and is thus difficult to be applied.

---

### Official Review · Reviewer_fhRF · 2025-10-27

**Soundness:** 3
**Presentation:** 3
**Contribution:** 2
**Rating:** 2
**Confidence:** 4

**Summary:**

The paper proposes MADRIVE, a memory-augmented reconstruction framework that enhances 3D Gaussian Splatting–based scene reconstructions for autonomous driving by replacing observed vehicles with visually similar 3D assets retrieved from a large external memory bank. The authors release MAD-CARS (~70K in-the-wild 360° car videos) and present a retrieval-and-reconstruction pipeline that finds nearest car instances, reconstructs their 3D geometry from video, and integrates them into target scenes via orientation alignment and relighting.

**Strengths:**

1. The paper is well written, with clear structure and professional presentation.

2. Figures and tables effectively communicate the method and results; visualizations are clear and informative.

3. The release of MAD-CARS is a meaningful community contribution that should facilitate reproducible research, stronger baselines, and broader progress in scene editing and AD simulation.

**Weaknesses:**

1. Comparative fairness. Since the method replaces in-scene vehicles with assets from a memory bank, direct comparisons against pure 3DGS baselines are not fully comparable. Please add evaluations against object-replacement or editing baselines (e.g., Chatsim [1]) under matched settings to establish fair performance gaps.

2. Quality and evaluation breadth. The presented results appear modest in fidelity (notably low resolution), lack vehicle rotation cases, and omit downstream AD task evaluations (e.g., perception/planning). Consider higher-resolution synthesis, sequences with significant yaw/pitch changes, and task-level tests (e.g., detector robustness, tracking consistency) to substantiate practical value.

3. Relighting effectiveness. In Figure 7, the visual gains from relighting are limited, and LPIPS differences at the reported scale (~0.3) are small.

Reference
[1] Wei Y, Wang Z, Lu Y, et al. Editable scene simulation for autonomous driving via LLM-agent collaboration. The First Workshop on Populating Empty Cities—Virtual Humans for Robotics and Autonomous Driving at CVPR 2024, 2nd Round. 2024.

**Questions:**

See the weaknesses.

---

### Official Review · Reviewer_gukj · 2025-10-28

**Soundness:** 3
**Presentation:** 3
**Contribution:** 3
**Rating:** 4
**Confidence:** 4

**Summary:**

This paper introduces MADRIVE, a framework for dynamic driving scene modeling that addresses a key limitation of existing reconstruction methods: the inability to photorealistically render vehicles from viewpoints not observed during training. The core idea is to replace partially-observed vehicles in a driving scene with high-fidelity, fully-reconstructed 3D assets retrieved from a large-scale external memory bank.

To enable this, the authors make two primary contributions. First, they present MAD-CARS, a new, large-scale dataset of approximately 70,000 360° multi-view videos of real-world cars, which is a significant contribution in its own right. Second, they propose a complete pipeline that:

1. **Retrieves** the most visually similar car from the MAD-CARS database using a combination of image embeddings and color attributes.
2. **Reconstructs** a relightable 3D asset of the retrieved car using 2D Gaussian Splatting, augmented with physically-based rendering (PBR) properties (albedo, roughness, metallicity) and several regularization techniques.
3. **Composes** the reconstructed asset into the target scene through careful alignment, relighting to match the scene's illumination, and shadow placement.

The authors introduce a challenging evaluation protocol focused on scene *extrapolation*—predicting future frames of a maneuver from past views. Experiments show that MADRIVE significantly outperforms state-of-the-art baselines in generating coherent and realistic future views, as measured by the performance of downstream perception tasks like tracking and segmentation on the synthesized images.

**Strengths:**

1. **Major Dataset Contribution:** The MAD-CARS dataset is a substantial contribution to the community. With ~70k instances, it dramatically expands the scale and diversity of publicly available multi-view car data. The detailed curation process described in the appendix further enhances its value and potential for future research in 3D reconstruction, novel view synthesis, and generative modeling.
2. **Strong and Comprehensive Technical Execution:** The proposed MADRIVE framework is technically sound and well-engineered. The pipeline is thoughtfully designed, integrating state-of-the-art techniques for retrieval (SigLIP2), reconstruction (Gaussian Splatting), and rendering (PBR-based relighting). The reconstruction module's use of opacity and normal regularization, along with synthetic data augmentation for disentangling lighting, demonstrates a sophisticated understanding of the challenges involved.
3. **Thorough and Meaningful Evaluation:** The evaluation methodology is a key strength.
    - The proposed *scene extrapolation* task is more challenging and practical than the standard interpolation/novel-view-synthesis tasks, providing a much stronger test of a model's generalization capabilities.
    - Using downstream perception metrics (MOTA, IDF1, Segmentation IoU) instead of just pixel-level metrics (like PSNR/SSIM) is an excellent choice. It evaluates the functional realism of the generated scenes—i.e., whether they are good enough for perception models, which is the ultimate goal in AD simulation.
    - The paper includes extensive qualitative results and detailed ablation studies (Fig 6, 7, 8) that convincingly justify the design choices for each component of the framework.

**Weaknesses:**

1. **Loss of Object Identity:** The most fundamental limitation is the trade-off between photorealism and identity preservation. The framework replaces an object with a *similar* one, not the *exact* one. This means any unique characteristics of the original vehicle (e.g., license plates, dents, scratches, bumper stickers, specific dirt patterns) are lost. For applications like "replaying" a safety-critical event for debugging, this loss of identity could be a critical flaw. The paper acknowledges this trade-off, but the implications could be discussed more deeply.
2. **Dependency and Scalability of the Memory Bank:** The success of MADRIVE is intrinsically tied to the comprehensiveness of the MAD-CARS database. The system's performance may degrade significantly if it encounters an out-of-distribution vehicle (e.g., a very rare model, a newly released car, or one with unusual modifications) that has no close match in the database. The framework is also currently specialized for cars; extending it to other dynamic actors like trucks, buses, or pedestrians would require creating similarly large-scale datasets for each category.
3. **Computational Complexity:** The end-to-end pipeline appears computationally expensive. For each dynamic vehicle in a scene, the system must perform a database query, followed by a full 3D reconstruction from a video sequence, and then a relighting and composition step. The paper does not provide an end-to-end runtime analysis, making it difficult to assess the practicality of this approach for generating large-scale simulations or for any near-real-time application.
4. **Potential for Imperfect Integration:** While the results are impressive, the integration of external assets into a scene is an extremely challenging problem. Subtle inconsistencies in lighting, shadows, or pose alignment can break the realism. The reliance on external models for environment map estimation and orientation refinement introduces potential points of failure.
5. **Limited Generalizability to Long-Tail and Atypical Object Categories:** The framework's demonstrated success is on passenger cars, a relatively common and homogenous category of road users. Its applicability to "long-tail" objects—such as large trucks, construction vehicles, emergency vehicles, or other atypical vehicle types—is questionable. The retrieval-based core is predicated on the existence of a close match within the memory bank. For a rare object not well-represented in MAD-CARS, the retrieval will likely return a semantically inappropriate substitute (e.g., a large SUV for a semi-truck), which would drastically break the scene's realism. In such cases, an in-situ reconstruction method, while perhaps less detailed on unseen parts, would at least preserve the object's correct class and general shape, which may be preferable. This raises questions about the scalability of the memory-augmented approach to capture the full diversity of real-world traffic.

**Questions:**

1. **On Identity vs. Realism:** The core concept sacrifices object identity for novel-view photorealism. Could you elaborate on the scenarios where this trade-off is acceptable versus those where it is not? Have you considered hybrid approaches that might use the retrieved asset as a geometric and textural prior but then try to "in-paint" or transfer specific details from the original observed views to the final model?
2. **Handling OOD and Non-Car Objects:** How does the system currently behave when a query vehicle has no close match in MAD-CARS? Does it fail gracefully or retrieve a poor match, leading to significant visual artifacts? What are your thoughts on extending this memory-augmented paradigm to other object categories like trucks or cyclists, and what would be the primary challenges in doing so?
3. **Runtime and Performance:** Could you please provide a breakdown of the computational cost and end-to-end processing time for a typical scene? For instance, how long does the retrieval, per-car reconstruction, and final scene composition take? Is the per-car reconstruction step pre-computed for the entire database, or is it performed on-the-fly after retrieval?
4. **Sensitivity of Relighting and Alignment:** The relighting process depends on an estimated environment map from `DiffusionLight`. How sensitive is the final rendered appearance to the accuracy of this estimated map, especially in complex lighting such as night scenes or sunsets? Similarly, how robust is the ICP-based alignment, and could small alignment errors be a contributing factor to the slightly lower MOTP score compared to Street-GS, as noted in Table 1?
5. **Comparison with Generative Approaches:** The paper rightly compares against reconstruction-based methods. However, another alternative could be to use a generative model (e.g., a diffusion model) conditioned on the observed views to complete the vehicle. While your ablation in Section G is helpful, could you provide a more in-depth discussion on the pros and cons of your retrieval-based approach versus a purely generative completion approach in the main paper?

---

### Official Review · Reviewer_hwNm · 2025-10-30

**Soundness:** 3
**Presentation:** 3
**Contribution:** 3
**Rating:** 6
**Confidence:** 3

**Summary:**

The paper introduces MADRIVE, a memory-augmented driving-scene reconstruction framework that replaces partially observed, dynamic vehicles with visually similar, relightable 3D assets retrieved from a large external memory bank (MAD-CARS).  The paper also introduces MAD-CARS, a curated ∼70k multi-view car-video dataset with brand/model/type/color diversity, and includes ablations on reconstruction regularizers, synthetic multi-illumination augmentation, retrieval quality (vs. 3DRealCar), and relighting.

**Strengths:**

1. Retrieval + relightable asset insertion is a pragmatic route to full 360° vehicle coverage from sparse in-scene views.
2. Physically based relighting integrated with 2D Gaussian splats, enabling plausible appearance under new illumination without multi-illumination capture.
3. Consistent gains in MOTA/IDF1 and segmentation IoU on synthesized future frames, with informative qualitative results.
4. MAD-CARS (∼70k car videos) enables better retrieval and more realistic assets than smaller real-car or CAD collections; the paper includes retrieval accuracy analyses and ablations.

**Weaknesses:**

1. Test-time insertion uses ground-truth 3D boxes; robustness to noisy boxes or tracker outputs is not reported. Provide stress tests that perturb box positions/orientations and quantify impacts on all metrics.

2. While color filtering helps, mis-retrievals (wrong trim level, body kit, or subtle geometry) could degrade realism. Add user studies or automatic metrics for fine-grained make/model/color matching on held-out scenes; report failure cases and a fallback strategy (e.g., top-k retrieval with consistency checks).

3. Environment maps are inferred from limited FOV, and shadowing uses a simple plane; quantify color-tone errors  and compare with learned/shadow-aware relighting or sun-position-aware methods.

4. Results are on 12 scenes; include more scenes or complementary datasets (e.g., NuScenes, KITTI-360) and report variance/confidence intervals.

5. Beyond Street-GS/HUGS/AutoSplat, consider dynamic-scene diffusion-NVS or recent 4D-GS variants; include a baseline that uses 3DRealCar or CAD insertion with the same relighting to isolate the benefit of retrieval on MAD-CARS.

6. Report end-to-end compute: retrieval latency, reconstruction time per asset, relighting/insertion cost, and rendering FPS. Provide guidelines for memory size vs. quality trade-offs and ablate top-k retrieval size.

**Questions:**

Please see Weaknesses.

---

### Note · Authors · 2025-11-13

**Comment:**

We thank all reviewers for their effort. We decided to withdraw our submission to make improvements.

**Withdrawal Confirmation:**

I have read and agree with the venue's withdrawal policy on behalf of myself and my co-authors.